# The governmental ranking of class and the academic performance of Indian adolescents

**Roshin Kunnel John**[1]*, **Boby Xavier**[1], **Anja Waldmeier**[1], **Andrea Hans Meyer**[2], **Jens Gaab**[1]

1 Division of Clinical Psychology and Psychotherapy, Department of Psychology, University of Basel, Basel, Switzerland, 2 Division of Developmental and Personality Psychology, Department of Psychology, University of Basel, Basel, Switzerland

* roshin.kunneljohn@unibas.ch

**Data Availability Statement:** All relevant data are in the paper and its Supporting Information file.

**Funding:** The authors received no specific funding for this work

## Abstract

Social and economic factors are commonly examined as contextual variables that predict academic achievement, apart from the educational environment. In India, a major segment of the socioeconomic status of students comprises the governmental stratification of population into three broad classes, viz., scheduled castes/tribes (SC-ST), other backward classes (OBC) and general class (GC). In this study, we examined the association of these governmental classes with the academic performance of Indian adolescents who enjoy the same school environment. Psychological measures of self-esteem and life satisfaction as well as demographic variables such as gender, age and family income were also examined as covariates. The study was conducted on a convenient sample of 858 students of X and XI grades. Based on multilevel regression models, the relationship between governmental classes and academic performance was significantly positive, wherein higher level of class predicted better academic performance. The study highlighted that students from the same school environment performed differently based on their social status and that this difference was not a function of their family income, thus pointing to potential role of non-economic aspects of the governmental stratification including caste affiliation. The findings indicate the need for further examining as well as planning to improve the aspects of students' social status that impact academic performance.

## Introduction

India's educational system has the context of a complex social fabric of casteist, religious, and regional diversities and hierarchies. Although the Indian constitution of 1950 eradicated the caste system, inequalities based on caste by birth has continued to hinder the national development [1, 2]. The Indian caste system is a religion-based hierarchical social structure, which divides the society into groups based on its members' occupations [3, 4]. With the aim of uplifting the disadvantaged groups, the government of India has grouped the traditional caste system into three classes, i.e., General Class (GC), Other Backward Classes (OBC) and Scheduled Castes/Scheduled Tribes (SC/ST). The major differences between caste and class are that

**Competing interests:** The authors have declared that no competing interests exist.

the membership in the caste is given by birth and that caste is a closed group characterized by endogamy whereas class is an open group. Also, in the class system vertical mobility is possible, wherein a person can move higher and go down, whereas in caste there is no such mobility. Finally, a given class can be distinguished from another class on the basis of economic criteria such as income and occupation whereas caste is based on religious and mythical traditions and traditional occupations [5]. But although the governmental classes are thought to replace castes and thus to eradicate caste-related discrimination, they still represent basically the caste system, since the class assignment is based on sub-caste affiliation, as opposed to individual socio-economic status [6, 7]. Thus, in the system of the governmental class, the so-called 'untouchables' (Dalits) are assigned to SC/ST (all abbreviations are explained in Table 1), other socio-economically unprivileged castes, such as 'shudras' are grouped into OBC while members of the highest caste being assigned to General or Forward class [6, 8]. Although the right to free and compulsory education for all children aged under 14 years is constitutionally guaranteed and was strongly advocated in the Right to Education Act [9], substantial inequality in education and employment still exists, and family income is strongly influenced by caste and ethnicity [10]. There are studies which show that the caste system contributes to economic inequality [11, 12]. Students belonging to the backward castes are exposed to various forms of daily humiliation, exploitation and exclusion in the schools [2, 13, 14].

According to the National Sample Survey (NSS 2011–12, 68[th] Round), the proportion of persons of SC-ST in India is 27.5%, OBC 44% and GC 28.5% [15]. Based on the yearly report of Ministry of Human Resource Development [16], at Senior Secondary school, out of the 24.7% students who got enrolled, 5.9% were from the SC-ST category, which is slightly less than their national the proportion (27.5%) in India. Similarly, out of 19.4 million students who enrolled for secondary school board examinations in 2016, 4.9 million were from SC-ST [16]. As per NSS 2011–12, the rate of educational attendance in India in the age range of 5–14 years were 88–89% for SC-ST, 90% for OBC, and 93% for GC [15]. For the age range of 15–19 years, the rate of attendance dropped and the gap between the three categories widened to 54–57% for SC-ST, 64% for OBC, and 71% for GC [15]. Dropout rates are higher for students from the SC-ST categories (19.4% for SC and 24.7% for ST) as compared with dropout rates for all students (17.1%) in India [16].

The impact of socioeconomic status on educational outcomes is an important concern in educational research [17–19]. Studies have consistently shown that socioeconomic

**Table 1. Abbreviations.**

| GC | General Class |
|---|---|
| OBC | Other Backward Class |
| SC | Scheduled Caste |
| ST | Scheduled Tribes |
| NSS | National Sample Survey |
| ESAG | Educational Statistics at A Glance |
| KL | Kerala |
| MP | Madhya Pradesh |
| AP | Academic Performance |
| govt. | Governmental |
| SES | Self-Esteem Scale |
| SWLS | Satisfaction with Life Scale |
| CBSE | Central Board of Secondary Education |
| SBE | State Board of Education |

status of parents as well as family distress significantly influences students' overall academic achievement [20, 21]. Sirin [17] conducted a meta-analysis of studies on the relationship between academic achievement and socioeconomic status, and included 74 independent studies comprising 101157 students and 6871 schools. A strong positive relation was found between socioeconomic status and academic achievement. School location and social status as minority were found to be the major influences on this positive relation.

Bourdieu [22] proposed an elaborate theory of social, economical, and cultural capitals. According to him possessing economic resources (economic capital) contributes to increased social connection and influence (social capital). In turn, social capital may facilitate increased educational opportunities/achievement, which is part of the cultural capital [22, 23]. In the Indian context, caste affiliation along with the financial condition of the family constitutes a major part of students' socioeconomic and cultural capital [24, 25]. Caste affiliation may determine to some extent the socioeconomic status of the family [11]. Also, economic status irrespective of caste/class affiliation may determine academic achievement [17, 19]. Gupta [26] found evidence for a differential influence of social status and economic status on academic performance in Indian college students. While students from lower castes were more likely to perform poorly in academics, students from lower economic status were not equally more likely to perform poorly.

According to the ESAG 2018 report [16], the proportion of students who passed the secondary school board examinations in 2016 was lower for the SC-ST categories of students (73% for SC and 65% for ST) when compared to all students (78.7%). Though this document does not mention GC and OBC, nonetheless, this information could be indicative of a trend that students from lower classes perform poorly at secondary schools across students of Central and State boards. Similar trend is observed also in the NSS 2011–12 [15]. In separate estimates for the states, in July 2012, a relatively lower rate of persons from the lower classes completed secondary education in Madhya Pradesh (SC-ST = 7%, OBC = 10%, GC = 15%), as well as Kerala (SC-ST = 15%, OBC = 18%, GC = 23%) [15]. A few Indian studies have addressed the role of caste status on academic achievement (e.g., [27–31]. Yadav and Chahal [27] observed that there was no significant difference of academic achievement between high and low caste students of secondary school. Sinha and Mishra [28] observed that social class-based identities especially linked to parental education did not determine academic achievement of Indian students. However, other studies observed that educational and occupational status of parents influence their academic achievement (e.g., [24, 32]). Whereas Sekhri [31] found that integrated college environment of higher and lower castes was unhelpful for academic achievement of both groups, Bagde et al [29] affirm that studying together did not have a negative impact on academic performance.

Also, self-esteem and life satisfaction have been found to impact academic performance [33–36]. There is evidence for a reciprocal association between self-esteem and academic achievement [37] as well as life satisfaction and academic achievement [38]. Indian studies have examined self-esteem and life satisfaction in the context of educational outcomes [12, 37]. However, there is a lack of research linking these variables to the governmental stratification of the three classes in the context of academic performance. Besides, the association between academic performance of Indian adolescent students and their governmental class has not been examined in any published study.

We therefore set out to investigate the association between governmental class and academic performance in Indian school students controlling for demographic variables such as family income, gender, and age, and psychological variables of self-esteem and life satisfaction.

## Materials and methods

We used across-sectional study design to examine the association between governmental class and academic performance. Based on the review of literature, we assumed that the governmental class would be associated with academic performance and that family income, self-esteem and life satisfaction would be covariates.

### Participants

The study was conducted on a sample of 858 students from the states of Kerala in South India, and Madhya Pradesh, in North India. Kerala ranks high among Indian states on social developmental and quality of life indicators, while Madhya Pradesh is close to India's average rates [39]. Kerala has the highest literacy rate (94%) among Indian states, higher than Madhya Pradesh (70%) as well as the national literacy rate (74%). Besides, Kerala boasts of equal educational opportunity for male and female children as compared to the other Indian states including Madhya Pradesh where females lag behind [39]. The participants were in the age range of 15–18 years, Mean age = 16.45 (and SD 0.78). There were 405 male and 453 female participants. The detailed description of the study sample is given in Table 2.

The participants were recruited from the X or XI grade students from six schools. From each randomly selected division of grade X or XI in each school, all the students in the division were included, which would minimize selection bias. The participating schools from the state of Madhya Pradesh followed CBSE syllabus (Central Board of Secondary Education) and the schools from Kerala followed State Board of Education (SBE) syllabus. The XI grade students were recruited from two different streams, i.e. science and commerce. All the schools were from the private sector and English was their medium of instruction. These schools were selected with the aim of incorporating urban and semi-urban population where all the three governmental classes are relatively more likely to be represented in a school [15]. For the same reason, either rural or metropolitan schools were not included. However, the three governmental classes were not proportionately distributed across the schools (Table 2). The lowest class (Scheduled Castes/Scheduled Tribes, SC/ST) was poorly represented in some schools. In the overall sample, there was a higher representation of General Class (GC) (i.e., 49%, with a proportion of 28.5% in India) and lower representation of SC-ST (i.e., 8% of our sample, as compared with a proportion of 27.5% in India). However, the percentage of Other Backward Classes (OBC) in our sample (i.e., 43%) was close to their actual proportion of 44% in India [15, 40]. According to NSS 2011–12 [15], Kerala and Madhya Pradesh differ regarding the proportion of the three govt. classes. Kerala has only 9.5% persons of SC-ST, 65% OBC, and 25% GC, compared to 39% SC-ST, 42% OBC, and 19% GC in Madhya Pradesh. Besides, in Madhya Pradesh, there is a lower rate of persons who complete secondary education as well as lower gap across the three categories (SC-ST = 7%, OBC = 10%, GC = 15%), as compared with Kerala (SC-ST = 15%, OBC = 18%, GC = 23%). Thus, our sample had a lower representation of the

**Table 2. Participants' demographic information: Frequencies/percentages.**

|  |  | Full Sample N/% | Individual Schools N/% |  |  |  |  |  |
|---|---|---|---|---|---|---|---|---|
|  |  |  | KL1: 105/12.2 | KL2: 236/27.5 | KL3: 247/28.8 | MP1: 123/14.3 | MP2: 25/2.9 | MP3: 122/14.2 |
| Gender | Male | 405/47.2 | 57/54.3 | 123/52.1 | 139/56.3 | 0/0 | 23/92 | 63/51.6 |
|  | Female | 453/52.8 | 48/45.7 | 113/47.9 | 108/43.7 | 123/100 | 2/8 | 59/48.4 |
| Govt. Class | SC-ST | 68/7.9 | 1/1.0 | 39/16.5 | 12/4.9 | 4/3.2 | 0/0 | 12/9.8 |
|  | OBC | 370/43.1 | 84/80.0 | 93/39.4 | 168/68.0 | 12/9.8 | 6/24 | 7/5.8 |
|  | GC | 420/49.0 | 20/19.0 | 104/44.1 | 67/27.1 | 107/87.0 | 19/76 | 103/84.4 |

lowest class (SC-ST) because their proportion is less than 10% in Kerala and only 7% of the SC-ST studied till Secondary School in Madhya Pradesh [15].

## Measures

The students' governmental class was obtained from the school registers, which record this to allow class-based reservation quotas. Students' level of academic performance was obtained from the exam-results from the school authorities. To get comparable results, grades as well as percentages of achieved points were transformed into z-scores. Individual mean z-score across all exams for each student was calculated as indicator of academic performance. Parents' monthly income was used as a proxy for socioeconomic status. The income of an Indian middle-class family with 2–3 earning adults broadly ranges between Rs 20000 to Rs 50000 [41, 42]. Students reported parents' monthly income on a five-point measure ranging from 1 to 5, where 1 = <5000 Rupees per month (which correspond to the lower class family income); 2 = 5001–20000 (lower middle class); 3 = 20001–50000 (middle class); 4 = 50001–100000 (upper middle class); and 5 = >100001 Rupees per month (upper class) [41, 42]. Self-esteem was assessed with the 10-item Rosenberg Self-Esteem Scale, a commonly used and well-validated measure of self-esteem [43, 44]. Responses were measured on a 4-point scale, ranging from 1 ("strongly disagree") to 4 ("strongly agree"). Satisfaction with Life Scale (SWLS) [45]. was used as a measure of participants' global cognitive judgment of life satisfaction. In this 5-item scale participants indicate how much they agree or disagree with each item on a 7-point scale that ranges from 7 ("strongly agree") to 1 ("strongly disagree").

## Procedure

The research project was submitted to the Cantonal Ethics Committee (Basel-Stadt and Basel-Land), which positively acknowledged the study protocol and informed consent forms, and approved the study. The necessary permissions from the respective school management trusties as well as the school principals in the states of Kerala and Madhya Pradesh were obtained and these were also submitted to the Cantonal Ethics Committee. Prior to data collection, written informed consent was obtained from all the participants in the age range of 17–18 years as well as from parents of all the participants in the age range of 15–16 years. Also, the written assent was obtained from all the participants in the age range of 15–16 years. The assessments were administered during school hours and in classrooms. Students were given 30 minutes to complete the assessment. The medium of assessment was English since all the participants were from English medium schools. Overall, 883 students were recruited. Of these, 25 had to be eliminated due to unknown governmental class.

## Statistical analyses

To test our hypothesis, we used multiple linear regression and multilevel models. Multiple linear regression models were used to assess school specific relationships between the factors governmental class and school performance, thereby controlling for student's sex, age, family income, self-esteem, and life satisfaction. Thus, separate regression models were run for each of the six schools. A multilevel model was then set up to assess the relationship between governmental class and school performance for all six schools combined, once again controlling for the above mentioned five covariates. This model contained a random intercept.

## Results

The average academic performance varied among the six schools (likelihood ratio = 230.0, p < .001) with values ranging between 37.6 (KL1) and 63.2 (MP1). Descriptives of the demographic variables and the measures for our sample are given in Tables 3 and 4. The intra-school correlation for academic performance was 0.32.

Multilevel analysis revealed significant differences in academic performance among the three governmental classes, when considering all schools together ($F_{2,845}$ = 5.73, p = 0.003). Predicted school performance values were 45.5 (±3.6), 49.1 (±3.3), and 51.4 (±3.3) for low, medium and high-class levels respectively, and were thus increasing with increasing levels of governmental class.

Assuming a linear functionality between governmental classes and academic performance, we obtained a positive association (β = 2.71, SE = 0.78, $t$ = 3.46, p<0.001), i.e. the higher the class level, the better was the academic performance (Table 5).

The association between covariates and academic performance, adjusted for governmental classes and each of the other covariates was as follows: self-esteem, β = 0.363, SE = 0.124, p = .004; life satisfaction β = 0.093, SE = 0.079, p = .244; gender, β = 8.294, SE = 1.001, p < .001 (females higher than males); family income, β = 0.360, SE = 0.525, p = .493; age, β = 0.031, SE = 0.618, p = .960.

Multiple regression analyses for the association between governmental class and academic performance of individual schools revealed the following pattern. Only in one school (KL2), there was a sizeable representation (See Table 2) of all the three govermental classes. As for the other 5 schools, there was no proportionate/adequate representation of all the three categories in each school when taken separately. Thus, when school KL2 was separately examined, it was found that academic performance was highest in the highest class (GC) and lowest in the lowest class (SC-ST) (p<0.001). Assuming a linear functionality between governmental class and academic performance, we found a positive association in this school (KL2: β = 5.12, SE = 1.23, $t$ = 4.17, p<0.001) where the students showed increasing academic performance with increasing class levels. For the other five schools, a significant pattern of association could not be observed (Table 6).

## Discussion

In this study, we examined the three governmental classes namely, SC-ST, OBC, and GC, with respect to their association with academic performance in Indian adolescent students of grades X and XI while controlling for age, gender, family income, self-esteem, and life satisfaction. Results based on multilevel regression analysis showed that lower class was likely to be associated with low academic performance. The association between social class and academic performance is a consistent finding in studies with ethnic minorities from low socio-economic status [17, 46–48]. For instance, in a meta-analytic review of research, Sirin [17] observed that

**Table 3. Descriptives: Academic performance, self-esteem and life satisfaction mean (SD).**

|  | AP | SES | SWLS |
|---|---|---|---|
| **Total:** N = 858 | 48.5 (16.2) | 18.1 (4.2) | 21.5 (6.0) |
| **Male:** n = 405 | 42.3 (15.7) | 18.2 (4.0) | 21.4 (5.8) |
| **Female:** n = 453 | 53.9 (14.5) | 17.9 (4.2) | 21.6 (6.2) |
| **SC-ST:** n = 68 | 44.5 (16.1) | 17.1 (3.3) | 21.5 (6.5) |
| **OBC:** n = 370 | 43.0 (14.7) | 19.1 (4.4) | 21.0 (5.9) |
| **GC:** n = 420 | 53.8 (15.7) | 17.3 (3.9) | 22.1 (6.0) |

**Table 4. Frequencies (percentages) of income and Govt.class.**

| Income | GC | OBC | SC-ST | Total |
|---|---|---|---|---|
| Below 5000 | 79(19) | 89(24) | 23 (34) | 191 |
| 5000–20000 | 137(33) | 179 (48) | 21 (31) | 337 |
| 2000–50000 | 135 (32) | 68 (18) | 16 (24) | 219 |
| 50000–1 lakh | 65 (16) | 22 (6) | 06 (9) | 93 |
| above 1 lakh | 04 (1) | 12 (3) | 02 (3) | 18 |
| Total | 420 | 370 | 68 | 858 |

minority students performed poorly as compared to their non-minority peers on account of important factors such as low family income, poor parental education and the influence of being in the neighborhood of low social status.

It is interesting to observe that the students' governmental class predicted their academic performance, while controlling for their family income, which itself had no influence on academic performance in our sample. In a study that examined how social and economic disadvantage influenced school performance, Considine and Zappalà [47] observed that the 'social' and the 'economic' components of the socioeconomic status may have distinct and separate influences on academic performance. Class-caste affiliation is an important aspect of the socio-economic status of Indian students. It is a complex mix of the 'social' and the 'economic', where the caste is more to do with social status and the class is to do with economic status. Broadly speaking, the three governmental classes may correspond to the higher (GC), the middle (OBC), and the lower social classes (SC-ST). However, the association between the three classes and family income is rather complex. In our sample, more students from the lower classes (especially the SC-ST) reported low family income (< 5,000 rupees or 5,000–20,000 rupees per month) when compared with the higher class (Table 4). But, a relatively higher proportion of the lower classes (especially OBC students), as compared to GC, reported very high family income (above 100,000 rupees per month). Thus, family income did not seem to correspond to social hierarchy at least in the case of very high-income families.

The reason for our results must be linked to the influence of social status over and above the potential influence of economic status (family income) as well as some of the psychological factors (self-esteem and general wellbeing / life satisfaction) which were controlled for and even though self-esteem itself was found to be a significant predictor of academic performance. Factors specific to belonging to a household of backward caste must have contributed to the poorer academic performance of the students from the backward castes/classes. Caste-based social identity is likely to be strengthened by caste-based social discrimination. For example, Nambeeshan [49] documented the discriminative experiences of the Dalit (SC-ST) students in

**Table 5. Regression coefficients of multilevel model with governmental class as continuous predictor (assuming a linear relationship) and academic performance as outcome, controlling for family income, gender, age, self-esteem, and life satisfaction.**

| Intercept | value | Std.error | DF | t-value | p-value |
|---|---|---|---|---|---|
| **Govt.class** | 2.71 | 0.78 | 846 | 3.46 | 0.006 |
| **SES** | 0.36 | 0.12 | 846 | 2.93 | 0.003 |
| **SWLS** | 0.09 | 0.08 | 846 | 1.16 | 0.244 |
| **Sex** | 8.29 | 1.00 | 846 | 0.28 | 0.001 |
| **Income** | 0.35 | 0.52 | 846 | 0.67 | 0.49 |
| **Age** | 0.02 | 0.61 | 846 | 0.04 | 0.96 |

**Table 6. Regression coefficients from linear regression models with governmental class as continuous predictor (assuming a linear relationship) and academic performance of individuals as outcome, controlling for family income, gender, age, self-esteem, and life satisfaction.** A separate model was run for each of the six schools.

| School name | Term | Estimate (β) | Std.error | statistic | p.value |
|---|---|---|---|---|---|
| KL1 | Govt. class | 0.15 | 2.68 | 0.05 | 0.96 |
| KL2 | Govt. class | 5.12 | 1.23 | 4.17 | 0.001 |
| KL3 | Govt. class | -1.21 | 1.68 | -0.72 | 0.47 |
| MP1 | Govt. class | 1.73 | 2.07 | 0.84 | 0.41 |
| MP2 | Govt. class | 6.02 | 7.71 | 0.78 | 0.44 |
| MP3 | Govt. class | 0.72 | 1.91 | 0.38 | 0.71 |

relation to a) access to school including facilities and resources b) participation in different spheres of school life and c) social relations with teachers and peers in schools in the state of Rajasthan. Discrimination can lead to decreased school engagement and rate of attendance as well as increased absenteeism and school dropout [49, 50]. Maurya [50] examined 13 Dalit narratives and found that Dalit students experienced exclusion and humiliation from class-mates, teachers and administration. An important theme that emerged in the narratives was 'learned self-devaluation', a tendency to devalue oneself as part of a group and be resigned to the inequalities/injustices imposed by others. Thus, it was observed that Dalit students would not ask for clarifications in a classroom or express their legitimate needs because of lack of assertiveness and self-confidence [50].

Students from the backward castes are likely to be low on educational and career aspirations because the household of the students may have less understanding of and exposure to educational and career opportunities different from their traditional caste-based social and occupational roles. For instance, according to NSS 2011–12 [15], one in four SC-ST households had no literate family member of age 15 years and above in rural area, as compared to one in ten households in urban area. On comparison, among OBC, 18% households in rural areas compared to 7 per cent in urban areas, and among GC, 11% households in rural areas compared to 3% in urban areas had no literate member of age 15 years and above [15]. The persons from the higher class were more likely to live and study in urban area as compared to the lower classes. Also, family members in the higher class were much more likely to have self-employment or salaried jobs as compared to the members of the lower classes who work mostly in the primary sector [15]. Thus, the role of (lack of) education, social exposure, career aspirations and achievement motivation in relation to the caste-based social identity must be examined in future studies.

According to NSS 2011–12 [15], in Kerala, there is a higher rate of persons who complete secondary education as well as lower gap across the three categories (SC-ST = 14–16%, OBC = 18%, GC = 23%), as compared with Madhya Pradesh (SC-ST = 5–8%, OBC = 10%, GC = 15%). Thus, although Kerala is relatively higher on social indicators [15, 51], the results for KL2 (Table 6) was similar to the overall sample. Hence, the difference among the classes across the states in India as well as the influence of the specific context of the states need to be also examined for a better understanding of the association of caste affiliation and academic performance.

The factors that influence academic performance may be diverse and complex for all the numerous subgroups placed under the three governmental classes. Some of the social subgroups under OBC category have high economic prosperity and respectable social status in some states/districts of India. In a study [27] conducted in the state of Haryana in which students from an economically advanced OBC subgroup were included, the academic performance of

these OBC students was found to be on par with the General Class. Hence, the differential influence of governmental class for students from socially backward families which are economically advanced versus economically backward needs to be examined in the Indian context.

Another important factor that may have influenced the students' academic performance could be English language proficiency since our participants were from schools with English as medium of instruction. Students from the lower classes are much more likely than their counterparts to come from families with parental lack of education and lack of proficiency in English language. This is likely to influence their academic performance. Also, availability and utilization of books to read at home and of tuition or special coaching at school, home or elsewhere need to be also explored as factors that may influence academic performance of students in the Indian context.

Findings of this study are based on data from only two Indian states and three schools from each, where all the students from the randomly selected division of the X or XI grade participated as a sample cluster. These schools were heterogenous with respect to the proportion of students from the three governmental classes who studied in the division of X or XI grade that was sampled. However, it needs to be noted that our sample points to the actual ground reality of the presence of students from these three social classes at the level of high school and higher secondary school in the private sector schools which generally provide better quality of education in India.

We examined students from the three governmental classes who study in the same school environment. We found that students who belong to the lower-class lag behind in their academic performance when compared to the students from the higher class even when they study together the same subjects in the same classroom. Palardy [52] observed that students from lower socioeconomic status are likely to be positively influenced when they study along with students of higher socioeconomic status and are likely to perform better academically.

## Conclusion and implications

In this study, we examined academic performance of Indian adolescents who studied in the same school environment but belonged to the three ranks of governmental class, SC-ST, OBC and GC. Multilevel regression indicated that higher class predicted better academic performance when controlled for age, sex, income, self-esteem, and life satisfaction. However, results based on separate student clusters from the schools were inconsistent. Though we used a sizable overall sample, there must be caution in generalizing the findings based only on a few schools, especially since the sample was heterogenous with respect to the representation of the three governmental classes in each school. Since we controlled school environment by selecting all students in a class-room, it is likely that other factors of the family and social environment may have contributed to the outcome, especially those linked to social status, such as parental education, neighborhood influence, and caste-related perceptions and experiences. Hence, future studies need to explore the factors and processes by which social status impacts academic performance.

## Supporting information

**S1 Data.**
(SAV)

## Author Contributions

**Conceptualization:** Roshin Kunnel John, Boby Xavier, Andrea Hans Meyer, Jens Gaab.

**Data curation:** Roshin Kunnel John, Boby Xavier, Anja Waldmeier, Andrea Hans Meyer, Jens Gaab.

**Formal analysis:** Roshin Kunnel John, Boby Xavier, Andrea Hans Meyer.

**Investigation:** Roshin Kunnel John, Anja Waldmeier, Jens Gaab.

**Methodology:** Roshin Kunnel John, Boby Xavier, Anja Waldmeier, Andrea Hans Meyer, Jens Gaab.

**Project administration:** Roshin Kunnel John, Boby Xavier, Anja Waldmeier.

**Resources:** Jens Gaab.

**Supervision:** Boby Xavier, Jens Gaab.

**Validation:** Roshin Kunnel John, Anja Waldmeier.

**Visualization:** Roshin Kunnel John, Anja Waldmeier, Jens Gaab.

**Writing – original draft:** Roshin Kunnel John, Andrea Hans Meyer, Jens Gaab.

**Writing – review & editing:** Roshin Kunnel John, Anja Waldmeier, Andrea Hans Meyer, Jens Gaab.

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
