## [Decision Letter · Decision Letter 0]

14 Apr 2020

PONE-D-19-31769

Governmental Ranking of Class and Academic Performance of Indian Adolescents

PLOS ONE

Dear Prof. Dr. Gaab,

Thank you for submitting your manuscript to PLOS ONE. After careful consideration, we
feel that it has merit but does not fully meet PLOS ONE’s publication criteria as it
currently stands. Therefore, we invite you to submit a revised version of the
manuscript that addresses the points raised during the review process.

I would like to thank you for your patience. Both reviewers liked your paper, and I
believe that their comments will help you improve the article. *In
addition to their feedback, I have the following
suggestions: *

p. 6 (Measures): You may wish to change the intervals, as in their
current form, they overlap (e.g., does a student with parents' income
20000 Rs belong to interval 2 or 3?).p. 7 (Statistical Analyses): You are using a random intercept model. Have
you considered also estimating a fixed effects (dummy variable)
model? On top of p. 9, you refer to Figure 1; however, the figure is not
provided. On the same page, the estimated Beta is 4.12, but the value in Table 5
reads 5.12. It would also be useful to explain all abbreviations in table notes (and
then referring to the first Table where the respective abbreviation was
explained). Furthermore, you may wish to use the same number of decimal
places in Tables 4 and 5. Is there a reason why you prefer two decimal
places in Table 4 and three (or even four) in Table 5? Values such as
0.000 or 4.263e-05 should be replaced by <0.001. In Table 5 you refer to the term "Gov.rank", which should probably be
"Govt.class". Also, please, check all your references carefully, as they appear to be
incomplete (and, e.g., ref 21 also contains JEL classification).

We would appreciate receiving your revised manuscript by May 29 2020 11:59PM. When
you are ready to submit your revision, log on to https://www.editorialmanager.com/pone/ and select the 'Submissions
Needing Revision' folder to locate your manuscript file.

If you would like to make changes to your financial disclosure, please include your
updated statement in your cover letter.

To enhance the reproducibility of your results, we recommend that if applicable you
deposit your laboratory protocols in protocols.io, where a protocol can be assigned
its own identifier (DOI) such that it can be cited independently in the future. For
instructions see: http://journals.plos.org/plosone/s/submission-guidelines#loc-laboratory-protocols

We look forward to receiving your revised manuscript.

Kind regards,

Tomáš Želinský, Ph.D.

Academic Editor

PLOS ONE

Journal Requirements:

1. Please provide additional details regarding participant consent. In the ethics
statement in the Methods and online submission information, please ensure that you
have specified (1) whether consent was informed and (2) what type you obtained (for
instance, written or verbal, and if verbal, how it was documented and witnessed). If
your study included minors, state whether you obtained consent from parents or
guardians. If the need for consent was waived by the ethics committee, please
include this information.

2. During our internal checks, the in-house editorial staff noted that you conducted
research or obtained samples in another country. Please check the relevant national
regulations and laws applying to foreign researchers and state whether you obtained
the required permits and approvals. Please address this in your ethics statement in
both the manuscript and submission information.

3. Thank you for your ethics statement : "The research project was submitted to the
Cantonal Ethics Committee (Basel-Stadt and Basel-Land), which positively
acknowledged the study protocol and informed consent forms, but which also required
the necessary permission from the respective school management trusties as well as
the permission of school principals, which was then ob-tained in subsequence. "

Please amend your current ethics statement to confirm that your named institutional
review board or ethics committee specifically approved this study.

5. Please upload a copy of Figure 1, to which you refer in your text on page 16. If
the figure is no longer to be included as part of the submission please remove all
reference to it within the text.

6. Please include a separate caption for each figure in your manuscript.

Reviewers' comments:

Reviewer's Responses to Questions

**Comments to the Author**

1. Is the manuscript technically sound, and do the data support the conclusions?

Reviewer #1: Yes

Reviewer #2: Partly

2. Has the statistical analysis been performed
appropriately and rigorously? 

Reviewer #1: Yes

Reviewer #2: Yes

3. Have the authors made all data underlying the
findings in their manuscript fully available?

Reviewer #1: Yes

Reviewer #2: Yes

4. Is the manuscript presented in an intelligible
fashion and written in standard English?

Reviewer #1: Yes

Reviewer #2: Yes

5. Review Comments to the Author

Reviewer #1: I found your paper very motivating and has global approach towards
issues on discrimination. A few more changes as suggested in the report will improve
the quality of the paper. Wish you all the best.

Reviewer #2: Review: PONE-D-19-31769

The manuscript deals with the relationship between the government classes and
academic performance (AP) of the students of class X and XI in two states of India.
The topic is of immense importance and quite under-researched. While the paper has
potential, it has not fully utilized the data that might have been available to the
authors. The literature survey needs to be broadened and application of literature
to the findings and discussion needs tightening. Following are my specific
comments:

What are the enrolment proportions and drop-outs in secondary and senior secondary
school for the government classes and their performance at the national level? See
MHRD report (ESAG).

Literature survey:

1. Bourdieu’s theory of ‘cultural capital’ is useful here to explain differences in
academic achievement based on socio-economic status.

2. Line: ‘Alternately, the associated socio-economic status may determine academic
achievement, irrespective of caste/class affiliation.’ Please give an
example/reference.

3. The authors might benefit from the following article that deals with the
relationship between caste and academic performance in higher education in
India:

Namrata Gupta (2019) Intersectionality of gender and caste in academic performance:
quantitative study of an elite Indian engineering institute, Gender, Technology and
Development, 23:2, 165-186. https://www.tandfonline.com/doi/abs/10.1080/09718524.2019.1636568

4. Some of the references are not specific and does not lead the reader to the
source. For instance: ‘India Go. Ministry of Social Justice and Empowerment,
Government of India 2018,Sep’. Which document are you referring to?

5. On Pg. 10: ‘Though Indian studies have examined self-esteem and life satisfaction
in the context of educational outcomes, …’ What Indian studies are you referring
to?

Methodology:

We need more information about schools. Are they government, private or mixed
schools? What is their language of instruction?

Were there any difficulties for the participants in filling up the questionnaire and
how were they handled? For instance, was the assessment administered in English? If
yes, could the students understand the language particularly in the self-esteem and
life satisfaction scales?

What is a z-score?

Results:

P. 14: ‘Predicted school performance values were 45.5 (±3.6), 49.1 (±3.3), and 51.4
(±3.3) for low, medium and high-class levels respectively, and were thus increasing
with increasing levels of governmental class.’ Please specify if these are mean
values taking all the schools together. Also in the column on AP, there does not
seem to be a linear relationship between class categories and AP, since, SC/ST
average is 44.5, OBC 43.0 and GC 53.8. Please explain. In Table 2, AP, SES and SWLS
numbers for various categories are out of what? AP seems to be percentage, but the
rest?

P. 16: Line: ‘Also, an increase of family income may not necessarily correspond to
better educational outcomes for the students of low social class. In other words,
the educational disadvantage that we observe in our results in the lower
governmental class, may not be equated with economic disadvantage.’ Please state the
reason for this.

P. 16: ‘In a study (40) conducted in the state of Haryana in which students from a
socioeconomically advanced OBC subgroup were included, the academic performance of
these OBC students was found to be on par with the General Class.’

This statement appears flawed as the Yadav and Chahal, the authors of the paper cited
here state in their own paper: ‘The reserved categories include OBC’s which is
socially backward but economically prosperous category.’ (p.36: Yadav and Chahal
2016). Hence, Yadav and Chahal seem to have studied OBCs that are socially backward
and cannot be referred to as ‘socioeconomically advanced’.

P. 14: Line: ‘In one school (KL2), academic performance was highest in the highest
class (GC) and lowest in the lowest class (SC-ST) (p<.001) while in another
school (KL3), academic performance also varied among classes, albeit being lowest
for the medium class (OBC) (p=0.014). For the remaining four schools, no significant
differences in academic performance among the three class levels were found…’

As your Table 1 shows, schools KL1 and MP2 have either a single or no SC/ST students.
Hence the claim that: ‘ For the remaining four schools, no significant differences
in academic performance among the three class levels were found…’ does not seem
pertinent. Instead it will be better to show the relationship between caste and AP
for GC and OBC for all the schools. The status of the OBCs in Kerala and MP should
also be discussed and related to the findings.

Discussion:

Based on the literature, the authors might like to discuss as to why there is a
relationship between government classes/caste and the AP.

Line: ‘(11) observed that students from lower socioeconomic status are likely to have
a positive influence...’

Do you mean to say ‘….likely to be positively influenced…’? Please replace (11) by
the last name of the referred author.

Line: ‘This would imply that the gap in academic performance between students from
higher and lower governmental classes are likely to widen if students from the
schools of the governmental sector and from the rural background are
investigated.’

Because??

6. PLOS authors have the option to publish the peer
review history of their article (what does this mean?). If published, this will
include your full peer review and any attached files.

If you choose “no”, your identity will remain anonymous but your review may still be
made public.

**Do you want your identity to be public for this peer review?** For
information about this choice, including consent withdrawal, please see our
Privacy Policy.

Reviewer #1: No

Reviewer #2: No

report.pdf
---

## [Author Response · Author response to Decision Letter 0]

29 May 2020

Response to Editor’s suggestions#1:

I would like to thank you for your patience. Both reviewers liked your paper, and I
believe that their comments will help you improve the article. In addition to their
feedback, I have the following suggestions: 

General author response to Editor’s comment

Thank you very much for your most helpful comments. We are really happy to receive
the comments from reviewers and additional suggestions from you. We have gone
through all the comments and have responded to each of them. In case any of these
responses need further modification, we would be most willing to do the same.

Comment 1

• p. 6 (Measures): You may wish to change the intervals, as in their current form,
they overlap (e.g., does a student with parents' income 20000 Rs belong to interval
2 or 3?).

Authors’ response to editor’s comment

Thank you for your valuable observation. In our revised manuscript, we have changed
the intervals to the following format.

1=<5000 Rupees per month 

2=5001-20000

3=20001-50000

4=50001-100000

5=100001-above

Thus, a student with Rs 20000 family income will be belong to the interval 2. 

Revision on Page 4, 

Students reported parents’ monthly income on a five-point measure ranging from 1 to
5, where 1 =<5000 Rupees per month (which correspond to the lower class family
income); 2=5001–20000 (lower middle class); 3=20001-50000 (middle class);
4=50001–100000 (upper middle class); and 5=>100001 Rupees per month (upper class)
(41, 42).

Comment 2

• p. 7 (Statistical Analyses): You are using a random intercept model. Have you
considered also estimating a fixed effect (dummy variable) model? 

Authors’ response to editor’s comment

Thank you very much for this suggestion. Yes, we also ran a fixed effects model, i.e.
a model in which schools were not considered random but fixed effects (where a
coefficient is estimated for each school). Results were comparable. 

However, we felt that the random effects model better corresponds to our study
design.

Comment 3

• On top of p. 9, you refer to Figure 1; however, the figure is not provided. 

Authors’ response to editor’s comment

Thank you very much for these specific observations. Figure 1 is dropped and we have
removed reference to the figure from the revised manuscript.

Comment 4

• On the same page, the estimated Beta is 4.12, but the value in Table 5 reads 5.12. 

Authors’ response to editor’s comment

Thank you very much for identifying this error. The estimated Beta is 5.12 as on
Table 5. Correction has been made in the revised manuscript.

Comment 5

• It would also be useful to explain all abbreviations in table notes (and then
referring to the first Table where the respective abbreviation was explained).
Furthermore, you may wish to use the same number of decimal places in Tables 4 and
5. Is there a reason why you prefer two decimal places in Table 4 and three (or even
four) in Table 5? Values such as 0.000 or 4.263e-05 should be replaced by <0.001. 

Authors’ response to editor’s comment

Thank you for this comment. We have placed all abbreviations in Table notes. Table 5
has been modified by using 2 decimal places as in Table 4 and errors such as 0.000
or 4.263e-05 have been corrected in the revised manuscript

Comment 6

• In Table 5 you refer to the term "Gov.rank", which should probably be "Govt.class". 

Authors’ response to editor’s comment

Thank you very much for pointing out this error. It has been corrected as ‘Govt.
class’ in the revised manuscript.

Comment 7

• Also, please, check all your references carefully, as they appear to be incomplete
(and, e.g., ref 21 also contains JEL classification).

Authors’ response to editor’s comment

Thank you very much for this comment. We have revised the reference section
thoroughly and made all the corrections.

Response to Additional Journal requirements #2:

1. Please provide additional details regarding participant consent. In the ethics
statement in the Methods and online submission information, please ensure that you
have specified (1) whether consent was informed and (2) what type you obtained (for
instance, written or verbal, and if verbal, how it was documented and witnessed). If
your study included minors, state whether you obtained consent from parents or
guardians. If the need for consent was waived by the ethics committee, please
include this information.

Authors’ response to the requirements

We have adhered to all these requirements and obtained written informed consent from
participants in the age range of 17-18 years as well as from parents of participants
in the age range of 15-16 years.

2. During our internal checks, the in-house editorial staff noted that you conducted
research or obtained samples in another country. Please check the relevant national
regulations and laws applying to foreign researchers and state whether you obtained
the required permits and approvals. Please address this in your ethics statement in
both the manuscript and submission information.

Authors’ response to the requirements

Thank you for your ethics statement : "The research project was submitted to the
Cantonal Ethics Committee (Basel-Stadt and Basel-Land), which positively
acknowledged the study protocol and informed consent forms, but which also required
the necessary permission from the respective school management trusties as well as
the permission of school principals, which was then ob-tained in subsequence. "

3. Please amend your current ethics statement to confirm that your named
institutional review board or ethics committee specifically approved this study.

Authors’ response to the requirements

We obtained the required permits and approvals for conducting research and obtaining
our sample in India. We have revised the manuscript by specifically mentioning the
Ethics committee approval.

Authors’ response to the requirements

The minimal anonymized data is uploaded as required

5. Please upload a copy of Figure 1, to which you refer in your text on page 16. If
the figure is no longer to be included as part of the submission please remove all
reference to it in the text.

6. Please include a separate caption for each figure in your manuscript.

Author response to reviewers’ comment

Figure 1 is dropped and we have removed reference to it from the revised manuscript.
There are no figures in the revised manuscript.

Response to Reviewer #1:

I found your paper very motivating and has global approach towards issues on
discrimination. A few more changes as suggested in the report will improve the
quality of the paper. Wish you all the best.

This paper studies the impact of caste system on academic performance of Indian
students in grades 10 and 11 who study in similar school environment. I found the
topic very interesting and relevant. Academic performance in these grades are very
important for Indian students and likely to affect the subjects they chose and the
schools/colleges they choose for further study. The results are also very thought
provoking and can have significant impact on policies directed towards backward
communities in India.

General author response to reviewers’ comment

Thank you very much for your most appreciative comments as well as for providing a
report to improve the quality of this paper. Thank you especially for highlighting
the strengths and scope of the study. We have reworked thoroughly on the manuscript
to incorporate the suggestions have pointed out. We have taken great care to adhere
and respond to each of them and revisied our manuscript accordingly. If you deem our
responses as not fully fitting or in case, we missed your point, we would of course
be willing to address them.

Comment 1

Describe the difference between caste system and government class in a bit more
detailed manner. The authors can either incorporate this description in the
introduction or put it in a footnote. The authors can also incorporate history on
caste system to explain why this system is so deep rooted in India even till this
day. Of course, apart from history there are other factors such as education that
has kept this system alive in India till this day. This will help the readers who
are not well aware of this difference.

Author response to reviewers’ comment

Thank you very much for this comment. We agree that our manuscript benefits from
including more information on the caste system and its relation to the governmental
classes. We have made the following additions in our manuscript in response to this
suggestion:

Revision in the manuscript on page 3: 

Although the Indian constitution of 1950 eradicated the caste system, inequalities
based on caste by birth has continued to hinder the national development (1. 2). The
Indian caste system can be defined as a system of social stratification, which
divides the society into groups based on its members occupations and is closely
associated with Hinduism (3,4). With the aim of uplifting the disadvantaged groups,
the government of India has grouped the traditional caste system into three classes,
i.e., General Class (GC), Other Backward Classes (OBC) and Scheduled
Castes/Scheduled Tribes (SC/ST). The major differences between caste and class are
that the membership in the caste is given by birth and that caste is a closed group
characterized by endogamy whereas class is an open group. Also, in the class system
vertical mobility is possible, such as a person can move higher and go down, whereas
in caste there is no such mobility. Finally, a given class can be distinguished from
another class on the basis of economic criteria such as income, occupation whereas
caste is based on religious and mythical traditions and may have hereditary and
traditional occupation (5). But although these governmental classes are thought to
replace castes and thus to eradicate caste-related discrimination, they still
represent basically the caste system, since class assignment is based on sub-caste
affiliation, as opposed to individual socioeconomic status (6, 7). Thus, in the
system of the governmental class, the so-called untouchables (Dalits) are assigned
to SC/ST, other socio-economically unprivileged castes, such as shudras are grouped
into OBC while members of the highest caste being assigned to General or Forward
class (6-8).

Comment 3

There are several examples of caste-based discrimination in India. It includes,
honour killing in case of inter-caste marriages, bullying students from lower castes
in higher educational institutions, untouchability, refusal to eat food cooked by
people belonging to lower castes etc. Using a wide array of examples from different
spheres of life in India will help readers understand the extent of discrimination
and the impact on the life of people belonging to lower castes due to this
discrimination.

Author response to reviewers’ comment

Thank you very much for these specific suggestions to improve the quality of our
introduction. As you have rightly mentioned, caste-based discrimination is a reality
in India and is manifest in various forms in different domain of life including
education. We have made a mention of this in the introduction in the following
line:

There are studies which show that the caste system contributes to economic inequality
(11, 12). Students belonging to the backward castes are exposed to various forms of
daily humiliation, exploitation and exclusion in the schools (2, 13,14).

The experience of discrimination is likely to impact students in their educational
outcome. However, while planning our paper, we decided to make a mention without
elaboration about the problem of caste-based discrimination in school along with
other important factors. This was based on the logic that the focus of the study
would be to dispassionately approach the class categories. Since the government
proposed to address the problem of caste by subsuming castes under the three
classes, the aim of our study was to examine whether the class division continues to
reflect the problem of caste system, by looking at academic performance. We would
prefer not to dilute this focus of the study by indulging in the larger issues
related to caste system such as caste discrimination and injustice especially in the
introduction. The results show that lower class has significantly lower academic
performance, and since economical and certain psychological factors have been
controlled for, the results point to the impact of social status, which is majorly
linked to caste-based social identity and experiences. In this context, we discuss
potential reasons related to social status that may have influenced the academic
performance. Here, in the revised manuscript caste discrimination is discussed as
one of the potential factors influencing academic performance. Provision of tuition
is another such factor that may influence academic outcome, as you have later
suggested. Our results provide the rationale for future studies to examine at length
the factors related to caste-based social status that may inflence academic
performance.

Revision on page 13 

Factors specific to belonging to a household of backward caste must have contributed
to the poorer academic performance of the students from the backward castes/classes.
Caste-based social identity may be strengthened by caste-based social
discrimination. For example, Nambeeshan (12) documents the discriminative
experiences of the Dalit students in relation to a) access to school including
facilities and resources b) participation in different spheres of school life and c)
Social relations with teachers and peers in schools in the State of Rajastan.
Discrimination can lead to decreased school engagement and rate of attendance as
well as increased absenteeism and school drop out (12, 13). Maurya (13) examined 13
Dalit narratives and found that Dalit students experienced exclusion and humiliation
from classmates, teachers and administration. An important theme that emerged in the
narratives was ‘learned self-devaluation’, a tendency to devalue onself as part of a
group and be resigned to the inequalities/injustices imposed by others. Thus, it was
observed that Dalit students would not ask for clarifications in a classroom or
express their legitimate needs because of lack of assertiveness and self-confidence
(13).

Comment 4 & Comment 7

Comment 4

Explain why choose Kerala and Madhya Pradesh. This is important and interesting since
Kerala has more liberal outlook to caste, religion etc. compared to Madhya Pradesh
which is very traditional.

Comment 7

Since Kerala and Madhya Pradesh are culturally quite different, it will be good if
the authors can control for state characteristics in their regressions. They can
either use state fixed effects if possible or can use data on caste-based crimes.
Such data can be found in the website of National Crime Records Bureau (India).
Also, the authors can run two separate regressions for the two states and compare
the results.

Author response to reviewers’ comment

Thank you very much for these important observations. As you have rightly pointed
out, the two States are interesting for their differences more than similarities. In
our revised manuscript we have incorporated information about the States of Kerala
and Madhya Pradesh and highlighted the relevant differences.

Revision on page 7

The study was conducted on school students from the states of Kerala in South India,
and Madhya Pradesh, in North India. Kerala ranks high among Indian states on social
developmental and quality of life indicators, while Madhya Pradesh is close to
India’s average rates (Census of India, 2011). Kerala has the highest literacy rate
(94%) among Indian states, higher than Madhya Pradesh (70%) as well as the national
literacy rate (74%). Besides, Kerala boasts of equal educational opportunity for
male and female children as compared to the other Indian states including Madhya
Pradesh where females lag behind (Census of India, 2011).

And again, on page 7

The participating schools from the State of Madhya Pradesh followed CBSE syllabus
(Central Board of Secondary Education) and Kerala schools followed State Board of
Education (SBE) syllabus.

And on page 8

According to NSS 2011-12 (15), Kerala and Madhya Pradesh differ regarding the
proportion of the three govt. classes. Kerala has only 9.5% persons of SC-ST, 65%
OBC, and 25% GC, compared to 39% SC-ST, 42% OBC, and 19% GC in Madhya Pradesh. In
Madhya Pradesh, there is a lower rate of persons who complete secondary education as
well as lower gap across the three categories (SC-ST = 7%, OBC = 10%, GC = 15%), as
compared with Kerala (SC-ST = 15%, OBC = 18%, GC = 23%) (15). Thus, our sample had a
lower representation of the lowest class (SC-ST) because their proportion is less
than 10% in Kerala and only 7% of the SC-ST studied till Secondary School in Madhya
Pradesh (15).

And on page 14

According to NSS 2011-12 (12), in Kerala, there is a higher rate of persons who
complete secondary education as well as lower gap across the three categories (SC-ST
= 14-16%, OBC = 18%, GC = 23%), as compared with Madhaya Pradesh (SC-ST = 5-8%, OBC
= 10%, GC = 15%). Thus, although Kerala is relatively higher on social indicators
(15, 51), the results for KL2 (Table 5) was similar to the overall sample.

However, in this study we have not sought to represent the States of Kerala and
Madhya Pradesh with our sample. We have used a purposive rather than random sample.
We have only examined data of 3 schools each in these States and only recruited
students from English medium schools of state board in Kerala and central board in
Madhya Pradesh. The design of the study is not one of comparative investigation of
the two states. Hence comparative analyses are beyond the scope of the study
objectives. Besides, without representative samples from the two states, comparative
regression models using either state fixed effects or crime data may not yield any
meaningful results.

Comment 5

In the methodology section, explain in brief the nature of assessment that students
were asked to do.

Author response to reviewers’ comment

Thank you for this comment. The assessment included the measures of self-esteem and
life satisfaction, self-report of family income in likert scale, and self-report of
gender and age and the governmental class and academic marks/grades were gathered
from school records as explained in the methodology section.

Comment 6

In the section statistical analysis, please check line 5 (in that section). I believe
a little rewording and rearranging the words can make the sentence construction
better.

Author response to reviewers’ comment

Thank you very much for this important observation. We have revised the manuscript
and corrected the sentence construction.

Revision on page 10

A multilevel model was then set up to assess the relationship between caste
affiliation school performance for all six schools combined, once again controlling
for the above mentioned five covariates.

Comment 7

Were the students asked if they take private tuition and if yes, then for how many
subjects? Indian students have a tendency to go for private tuition and that can
impact their academic performances.

Author response to reviewers’ comment

Thank you very much for this comment. We have not sought information on availability
of tuition/coaching at home or outside. Having tuition/coaching or having a family
member who can help in clarifying doubt are factors associated with the student’s
socio-economic status. As the reviewer has rightly observed, these factors need to
be explored in future research.

Revision on page 15

Also, availability and utilization of tuition or special coaching at school, home or
elsewhere need to be also explored as a factor that may influence academic
performance of students in the Indian context.

Comment 7

It would have been interesting if the authors could include data on students in the
arts and humanities stream. In India, there is a bias against students in
humanities.

Author response to reviewers’ comment

Thank you very much for this comment as well. Yes, it is true that there are more
students who go for science and commerce streams at higher secondary level and more
schools at private sector which provide these streams.

Response to Reviewer #2:

The manuscript deals with the relationship between the government classes and
academic performance (AP) of the students of class X and XI in two states of India.
The topic is of immense importance and quite under-researched. While the paper has
potential, it has not fully utilized the data that might have been available to the
authors. The literature survey needs to be broadened and application of literature
to the findings and discussion needs tightening. Following are my specific
comments

Response to the Comment

Thank you very much for your highly helpful comments. We have taken utmost care to
incorporate all your suggestions and to respond to all of observations. We have
revisied our manuscript accordingly. We have modified the introduction, method,
results and discussion on the basis of your comments. If you deem our responses as
not fully fitting or in case, we missed your point, we would of course be willing to
address them further.

Comment 1: Introduction

What are the enrolment proportions and drop-outs in secondary and senior secondary
school for the government classes and their performance at the national level? See
MHRD report (ESAG).

Authors’ response to reviewers’ comment

Thank you very much for the observation and for providing a relevant reference. This
information is definitely useful while discussing the difference in academic
performance across the governmental classes. However, we do not have data about the
proportions of enrollment, drop-outs and academic performance of all the three
governmental classes in the Ministry of Human Resource Developmentestimates (ESAG
2018). The report mentions theoverall data and then separate data for only SC-ST
categories. There are no separate data on the Other Backward Castes (OBCs) and
General Class. Based on the ESAG 2018, at Senior Secondary school, out of the 24.7
students who get enrolled, 5.9 are from the SC-ST category, which is slightly less
than their nationalproportion of the SC-ST category in India (27.5%). Similarly, out
of 19.4 million students who enrolled for secondary school board examinations in
2016, 4.9 million were from SC-ST. As per the 68th Round of the National Sample
Survey (NSS 2011-12), the rate of educational attendance in India in the age range
of 5-14years were 88-89% for SC-ST, 90% for OBC, and 93% for GC. For the age range
of 15-19 years, the rate of attendance dropped and the gap between the three
categories widened to 54-57% for SC-ST, 64% for OBC, and 71% for GC. Drop out rates
are higher for students from the SC-ST categories (19.4% for SC and 24.7% for ST) as
compared with drop out rates of all students (17.1%).More importantly, as per the
MHRD report the proportion ofstudents who passed the secondary school board
examinations in 2016 was lower for the SC-ST categories of students (73% for SC and
65% for ST) when compared to all students (78.7%). Though this document does not
mention GC and OBC, nonetheless, this information could be indicative of a trend
that students from lower classes perform poorly at secondary schools across students
of Central and State boards. Similar trend is observed also in the NSS 2011-12. In
separate estimates for the States, in July 2012 arelatively lower rate of persons
from the lower classes completed secondary education in Madhya Pradesh (SC-ST = 7%,
OBC = 10%, GC = 15%), as well as Kerala (SC-ST = 15%, OBC = 18%, GC = 23%).

Revision on page 4

According to the National Sample Survey (NSS 2011-12, 68th Round), the proportion of
persons SC-ST in India is 27.5%, OBC 44% and GC 28.5% (15). Based on the yearly
report of Ministry of Human Resource Development (ESAG 2018), at Senior Secondary
school, out of the 24.7 students who gott enrolled, 5.9 were from the SC-ST
category, which is slightly less than their national the proportion (27.5%) in India
(16). Similarly, out of 19.4 million students who enrolled for secondary school
board examinations in 2016, 4.9 million were from SC-ST. As per NSS 2011-12, the
rate of educational attendance in India in the age range of 5-14years were 88-89%
for SC-ST, 90% for OBC, and 93% for GC. For the age range of 15-19 years, the rate
of attendance dropped and the gap between the three categories widened to 54-57% for
SC-ST, 64% for OBC, and 71% for GC (15). Drop out rates are higher for students from
the SC-ST categories (19.4% for SC and 24.7% for ST) as compared with drop out rates
for all students (17.1%) in India (16).

Revision on page 5

According to the ESAG 2018 report (16), the proportion of students who passed the
secondary school board examinations in 2016 was lower for the SC-ST categories of
students (73% for SC and 65% for ST) when compared to all students (78.7%). Though
this document does not mention GC and OBC, nonetheless, this information could be
indicative of a trend that students from lower classes perform poorly at secondary
schools across students of Central and State boards. Similar trend is observed also
in the NSS 2011-12 (15). In separate estimates for the States, in July 2012 a
relatively lower rate of persons from the lower classes completed secondary
education in Madhya Pradesh (SC-ST = 7%, OBC = 10%, GC = 15%), as well as Kerala
(SC-ST = 15%, OBC = 18%, GC = 23%) (15).

References added:

National Sample Survey, 68th Round (July 2011 – June 2012). Employment and
Unemployment Situation among Social Groups in India. Ministry of Statistics and
ProgrammeImplementation, Government of India. 2015. http://mospi.nic.in/sites/default/files/national_data_bank/pdf/NSS_68Round-563.pdf

Educational Statistics at A Glance. Ministry of Human Resource Development,
Government of India. ESAG 2018. https://mhrd.gov.in/sites/upload_files/mhrd/files/statistics-new/ESAG-2018.pdf

Comment 2: Literature survey

Bourdieu’s theory of ‘cultural capital’ is useful here to explain differences in
academic achievement based on socio-economic status.

Authors’ response to reviewers’ comment

Thank you very much for your valuable suggestion. We have revised the discussion in
the manuscript to incorporate mention of how social and cultural capital is
transferable to economic capital and hence how the caste system can disadvantage the
students from lower castes economically.

Revision on page 5

Bourdieu (22) proposed an elaborate theory of social, economical, and cultural
capitals. According to him possessing economic resources (economic capital)
contributes to increased social connection and influence (social capital). In turn,
social capital may facilitate increased educational opportunities/achievement, which
is part of the cultural capital (22, 23). In the Indian context, caste affiliation
along with the financial condition of the family constitutes a major part of
students’ socioeconomic and cultural capital (24, 25). Caste affiliation may
determine to some extent the socioeconomic status of the family (11). Also, economic
status irrespective of caste/class affiliation may determine academic achievement
(17, 19). Gupta (26) found evidence for a differential influence of social status
and economic status on academic performance in Indian college students. While
students from lower castes were more likely to perform poorly in academics, students
from lower economic status were not more likely to perform poorly.

References added:

Bourdieu, P. The forms of capital. In J. Richardson (Ed.) Handbook of Theory and
Research for the Sociology of Education (New York, Greenwood). 1986: 241-258.

Huang, X. Understanding Bourdieu - Cultural Capital and Habitus. Rev. Eur. Studies.
2019: 11(3), 45-49. doi: 10.5539/res.v11n3p45

Comment 3: 

‘Alternately, the associated socio-economic status may determine academic
achievement, irrespective of caste/class affiliation.’ Please give an
example/reference.

Authors’ response to reviewers’ comment

This statement means that having higher/lower financial and social status may lead to
access to better/poorer schooling and additional coaching opportunities, and lead to
better/poorer academic achievement. As you have already suggested, the concept of
different forms of capitals (Bourdieu, 1986) and their relations can explain this.
The manuscript has been modified by providing references for this statement.

Comment 4: 

The authors might benefit from the following article that deals with the relationship
between caste and academic performance in higher education in India:

Namrata Gupta (2019) Intersectionality of gender and caste in academic performance:
quantitative study of an elite Indian engineering institute, Gender, Technology and
Development, 23:2, 165-186.https://www.tandfonline.com/doi/abs/10.1080/09718524.2019.1636568

Authors’ response to reviewers’ comment

Thank you very much for suggesting this reference to improve the literature survey.
The manuscript has been modified to incorporate reference to this article. The
referred study found that while students from lower castes were more likely to
perform poorly in academics, students from lower economic status were not more
likely to perform poorly.

Revision on page 5

Gupta (26) found evidence for a differential influence of social status and economic
status on academic performance in Indian college students. While students from lower
castes were more likely to perform poorly in academics, students from lower economic
status were not more likely to perform poorly.

Comment 5:

Some of the references are not specific and does not lead the reader to the source.
For instance: ‘India Go. Ministry of Social Justice and Empowerment, Government of
India 2018, Sep’. Which document are you referring to?

Authors’ response to reviewers’ comment

Thank you very much for bringing to notice this error in referencing. In the revised
manuscript, we have also gone through all references and ensured specificity as you
have suggested. 

The mentioned reference is corrected as: 

Social Justice AR 2018-19. Ministry of Social Justice and Empowerment, Government of
India 2018 http://socialjustice.nic.in/writereaddata/UploadFile/Social_Justice_AR_2018-19_English.pdf

Comment 6:

On Pg. 10: ‘Though Indian studies have examined self-esteem and life satisfaction in
the context of educational outcomes, …’ What Indian studies are you referring
to?

Authors’ response to reviewers’ comment

Thank you very much for the comment. In the revised manuscript we haveprovided the
references for this part of the statement. The references are (15) and (29).

Comment 7: Methods

We need more information about schools. Are they government, private or mixed
schools? What is their language of instruction?

Were there any difficulties for the participants in filling up the questionnaire and
how were they handled? For instance, was the assessment administered in English? If
yes, could the students understand the language particularly in the self-esteem and
life satisfaction scales?

What is a z-score?

Authors’ response to reviewers’ comment

Thank you very much for this comment. We have revised the methods section and the
suggested details regarding participants and procedure have been incorporated. Since
we conducted the study only in urban/semi-urban English medium schools, the
questionnaires were administered in English. Students were assisted by their
respective Eglish teachers in responding to the questionnaires.

The raw scores of academic performances from different streams of schools were
converted to z-scores for making them comparable.

Revised on page 7

The participating schools from the State of Madhya Pradesh followed CBSE syllabus
(Central Board of Secondary Education) and Kerala schools followed State Board of
Education (SBE) syllabus. The XI grade students were recruited from two different
streams, i.e. science and commerce. These schools were from the private sector and
English was the medium of instruction in all these schools.

Revised on page 9

The assessments were administered during school hours and in classrooms. Students
were given 30 minutes to complete the assessment. The medium of assessment was
English since all the participants were from English medium schools.

Comment 8a: Results

P. 14: ‘Predicted school performance values were 45.5 (±3.6), 49.1 (±3.3), and 51.4
(±3.3) for low, medium and high-class levels respectively, and were thus increasing
with increasing levels of governmental class.’ Please specify if these are mean
values taking all the schools together. Also, in the column on AP, there does not
seem to be a linear relationship between class categories and AP, since, SC/ST
average is 44.5, OBC 43.0 and GC 53.8. Please explain.

Author’s response to reviewers’ comment

We distinguish between AP mean values based on descriptive statistics (mean values
across all students and schools by caste-class) and predicted mean values based on a
multilevel model with caste taken as a factor with three levels (low, medium, and
high). In both cases, all schools were included to compute means.

In addition (and independent of the computation of means as explained above), we set
up an apriori hypothesis in which we wanted to test whether AP linearly increased
with higher caste class (from SC-ST to OBC to GC). To this end we used caste-class
as a linear predictor in a separate multilevel model.

Comment 8b

In Table 2, AP, SES and SWLS numbers for various categories are out of what? AP seems
to be percentage, but the rest?

Authors’ response to reviewers’ comment

Thank you for making these observations. These numbers are Mean (SD) as given on the
Table description on top [Table2: Descriptives: Academic performance, self-esteem
and life satisfaction Mean (SD)]

Comment 9: Discussion

P. 16: Line: ‘Also, an increase of family income may not necessarily correspond to
better educational outcomes for the students of low social class. In other words,
the educational disadvantage that we observe in our results in the lower
governmental class, may not be equated with economic disadvantage.’ Please state the
reason for this.

Authors’ response to reviewers’ comment

Thank you very much for making these careful observations. That social backwardness
may affect educational outcome independent of financial status is an argument that
is presented in the introduction with supportive references(page5). Further, the
results also point in this direction because caste affiliation contributed to
academic performance when family income was controlled for. Additionally, to support
this argument further, here we have added a descriptive observation that in our
sample a relatively higher proportion of OBC class reported higher family income
when compared with GC (This is only an additional observation and not a statistical
evidence). In the revised manuscript these two sentences (‘Also, an increase of
family income.... with economic disadvantage.’ have been since the argument is
discussed elsewhere.

Comment 10

P. 16: ‘In a study (40) conducted in the state of Haryana in which students from a
socioeconomically advanced OBC subgroup were included, the academic performance of
these OBC students was found to be on par with the General Class.’

This statement appears flawed as the Yadav and Chahal, the authors of the paper cited
here state in their own paper: ‘The reserved categories include OBC’s which is
socially backward but economically prosperous category.’ (p.36: Yadav and Chahal
2016). Hence, Yadav and Chahal seem to have studied OBCs that are socially backward
and cannot be referred to as ‘socioeconomically advanced’.

Authors’ response to reviewers’ comment

Thank you very much for carefullyidentifying this error. The statement has been
corrected as “economically backward”. The phrase “socially backward families which
are economically advanced” in the next sentence would further clarify it.

Comment 11

P. 14: Line: ‘In one school (KL2), academic performance was highest in the highest
class (GC) and lowest in the lowest class (SC-ST) (p<.001) while in another
school (KL3), academic performance also varied among classes, albeit being lowest
for the medium class (OBC) (p=0.014). For the remaining four schools, no significant
differences in academic performance among the three class levels were found…’

As your Table 1 shows, schools KL1 and MP2 have either a single or no SC/ST students.
Hence the claim that: ‘For the remaining four schools, no significant differences in
academic performance among the three class levels were found…’ does not seem
pertinent. Instead it will be better to show the relationship between caste and AP
for GC and OBC for all the schools. The status of the OBCs in Kerala and MP should
also be discussed and related to the findings.

Authors’ response to reviewers’ comment

Thank you very much for this observation. The main focus of the study is to look at
the overall sample (N=858) in the three categories. Examining of separate schools is
only additional and not part of the design or objectives of the study. Since we
planned to include the whole classroom while sample recruitment, the actual presence
of the three categories of classes in these schools are reflected in the sample.
Thus, KL1 and MP2 have either a single or no SC-ST students, and MP2has only 6 and
MP3 has only 7 OBC students. GC are relatively better represented in all six
schools. It may be interesting to notice that in KL2 where all three categories have
sizeable representation, academic performance was highest in the highest class (GC)
and lowest in the lowest class (SC-ST) (p<0.001), which is similar to our results
for the overall data. As you have suggested, we have modified the text by omitting
the statement ‘For the remaining schools...’ andour focusinstead will beon the three
categories for all schools. Besides, as suggested, we also elaborate on the status
of the govt classesin Kerala and MP.

Revised section on page 8

According to NSS 2011-12, 93% population in Kerala as compared to 72% in Madhya
Pradesh are literate. The two States also differ regarding the proportion of the
three govt. classes. Kerala has only 9.5% persons of SC-ST, 65% OBC, and 25% GC,
compared to 39% SC-ST, 42% OBC, and 19% GC in Madhya Pradesh. In Madhya Pradesh,
there is a lower rate of persons who complete secondary education as well as lower
gap across the three categories (SC-ST = 7%, OBC = 10%, GC = 15%), as compared with
Kerala (SC-ST = 15%, OBC = 18%, GC = 23%). Thus, our sample had a lower
representation of the lowest class (SC-ST) because their proportion is less than 10%
in Kerala and only 7% of the SC-ST studied till Secondary School in Madhya
Pradesh.

Revised section on page 11

Only in one school (KL2), there was a sizeable representation (See Table 1) of all
the three govt. classes. For the other 5 schools, there was no
proportionate/adequate representation of all three categories in each school when
taken separately. Thus, when school KL2 was separately examined, it was found that
academic performance was highest in the highest class (GC) and lowest in the lowest
class (SC-ST) (p<.001). Assuming a linear functionality between governmental
classes and academic performance, we found a positive association in this school
(KL2: β=4.12, SE=1.23, t=4.17, p<0.001)where the students showed increasing
academic performance with increasing class levels. For the other five schools, a
significant pattern of association could not be observed (Table 5).

Revised sections on page 14-15

For instance, according to NSS 2011-12 (15), one in four SC-ST households had no
literate family member of age 15 years and above in rural area, as compared to one
in ten households in urban area. On comparison, among OBC, 18% households in rural
areas compared to 7 per cent in urban areas and among GC, 11% in rural areas
compared to 3% in urban areas had no literate member of age 15 years and above (15).
The persons from the higher class were more likely to live and study in urban area
as compared to the lower classes. Also, family members in the higher class were much
more likely to have self-employment or salaried jobs as compared to the members of
the lower classes who work mostly in the primary sector (15). Thus, the role of
(lack of) education, social exposure, career aspirations and achievement motivation
in relation to the caste-based social identity must be examined in future studies. 

According to NSS 2011-12 (15), in Kerala, there is a higher rate of persons who
complete secondary education as well as lower gap across the three categories (SC-ST
= 14-16%, OBC = 18%, GC = 23%), as compared with Madhaya Pradesh (SC-ST = 5-8%, OBC
= 10%, GC = 15%). Thus, although Kerala is relatively higher on social indicators
(15, 51), the results for KL2 (Table 5) was similar to the overall sample. Hence,
the difference among the classes across the States in India as well as the influence
of the specific context of the States need to be also examined for a better
understanding of the association of caste affiliation and academic performance. 

Comment 12

Based on the literature, the authors might like to discuss as to why there is a
relationship between government classes/caste and the AP.

Authors’ response to reviewers’ comment

Thank you very much for the suggestion and our discussion has been modified to
incorporate it. In this study, we have not examined the factors that might have
contributed to the relationship between the classes and AP. Hence, after
corroborating our results by refering to other studies, we go on to make
hypothetical speculations on possible reasons for the relationshipin three
paragraphs.Following are someof the reasons that we have discussed in the original
manuscript:

1) The reason for our finding must be related to social status over and above the
potential influence of economic status (family income)as well as some of the
psychological factors (self-esteem and general wellbeing/life satisfaction) that
were controlled for. In other words, reasons specific to belonging to a household of
backward caste must have contributed to the poorer academic performance of the
students from the backward castes/classes. We leave it to future studies to examine
what aspects of caste identity and social status influence academic performance. In
our revised manuscript, we include that limited social exposure and minimal
aspiration/achievement motivation are likely to influence academic performance.

2) The influence of social backwardness on education may be increased by associated
economic backwardness. We refer to study which found that socially backward students
who were economically advanced performed well in academics on par with GC students
(The manuscript has been modified to bring emphasis on this argument). 

3) Yet another reason could be English language proficiency since our participants
were from schools with English as medium of instruction. Students from the lower
classes are much more likely to come from families with parental lack of education
and lack of proficiency in English language. This is likely to influence their
academic performance.

The manuscript has been modified to strengthen these arguments.

Revised section on page 14-15

The reason for our results must be linked to the influence of social status over and
above the potential influence of economic status (family income) as well as some of
the psychological factors (self-esteem and general wellbeing/life satisfaction) that
were controlled for. Factors specific to belonging to a household of backward caste
must have contributed to the poorer academic performance of the students from the
backward castes/classes. Students from the backward castes are likely to be low on
educational and career aspirations. This could be so because the household of the
students may have less understanding of or exposure to educational and career
opportunities different from their traditional caste-based social and occupational
roles. Hence, the potential role of caste-based social identity and its relation to
(lack of) social exposure, career aspirations and achievement motivation must be
examined in future studies. 

The factors that influence academic performance may be diverse and complex for all
the numerous subgroups placed under the three governmental classes. Some of the
social subgroups under OBC category have high economic prosperity and respectable
social status in some states/districts of India. In a study (27) conducted in the
state of Haryana in which students from an economically advanced OBC subgroup were
included, the academic performance of these OBC students was found to be on par with
the General Class. Thus, the influence of social backwardness on education may be
increased by associated economic backwardness. Hence, the differential influence of
governmental class for students from socially backward families which are
economically advanced versus economically backward needs to be examined in the
Indian context. 

Another important factor that may have influenced the students’ academic performance
reason could be English language proficiency since our participants were from
schools with English as medium of instruction. Students from the lower classes are
much more likely than their counterparts to come from families with parental lack of
education and lack of proficiency in English language. This is likely to influence
their academic performance.

Comment 13

‘(11) observed that students from lower socioeconomic status are likely to have a
positive influence...’

Do you mean to say ‘…. likely to be positively influenced…’? Please replace (11) by
the last name of the referred author.

Authors’ response to reviewers’ comment

Thank you for making this suggestion and pointing out the error. Yes. We meant to say
“... likely to be positively influenced”.Also, (11) is replaced with the author’s
last name.

Comment 14

‘This would imply that the gap in academic performance between students from higher
and lower governmental classes are likely to widen if students from the schools of
the governmental sector and from the rural background are investigated.’

Because??

Authors’ response to reviewers’ comment

We collected our data from urban areas where all the three class categories of
students are likely to study together in the same schools. This co-education is
likely to improve the academic performance of the lower class as observed in the
referred article. On the other hand, in rural areas and especially in schools of
government sector which provide free education, students from higher class are
seldom represented and hence there would not be possiblity for co-education of
higher and lower classes. Hence, we specualte that without the potential advantage
of co-education, these students are likely to perform poorer academically, if
compared with students of lower class who study with students of higher class. The
text has been modified to make this argument more lucid.

Revised text on page 16

Thomson (18) observed that students from lower socioeconomic status are likely to be
positively influenced when they study along with students of higher socioeconomic
status and are likely to perform better academically. Thus, if studying together
with students from higher governmental class is advantage for students from lower
class, then it would imply that we may find a greater gap in academic performance if
we include students from rural schools who are mostly from the lower classes and who
are less likely to study with students of higher class.

---

## [Decision Letter · Decision Letter 1]

28 Sep 2020

PONE-D-19-31769R1

Governmental Ranking of Class and Academic Performance of Indian Adolescents

PLOS ONE

Dear Dr. Gaab,

Thank you for submitting your manuscript to PLOS ONE. After careful consideration, we
feel that it has merit but does not fully meet PLOS ONE’s publication criteria as it
currently stands. Therefore, we invite you to submit a revised version of the
manuscript that addresses the points raised during the review process.

Once again, I would like to thank you for your patience. Reviewer #2 suggests two
minor points you might wish to consider in the discussion section. Once you address
these two minor comments (or explain why not include them), I’ll be happy to
recommend the paper for publication.

Please submit your revised manuscript by Nov 12 2020 11:59PM. If you will need more
time than this to complete your revisions, please reply to this message or contact
the journal office at plosone@plos.org. When
you're ready to submit your revision, log on to https://www.editorialmanager.com/pone/ and select the 'Submissions
Needing Revision' folder to locate your manuscript file.

If you would like to make changes to your financial disclosure, please include your
updated statement in your cover letter. Guidelines for resubmitting your figure
files are available below the reviewer comments at the end of this letter.

We look forward to receiving your revised manuscript.

Kind regards,

Tomáš Želinský, Ph.D.

Academic Editor

PLOS ONE

Reviewers' comments:

Reviewer's Responses to Questions

**Comments to the Author**

1. If the authors have adequately addressed your comments raised in a previous round
of review and you feel that this manuscript is now acceptable for publication, you
may indicate that here to bypass the “Comments to the Author” section, enter your
conflict of interest statement in the “Confidential to Editor” section, and submit
your "Accept" recommendation.

Reviewer #1: All comments have been addressed

Reviewer #2: All comments have been addressed

2. Is the manuscript technically sound, and do the data
support the conclusions?

Reviewer #1: Yes

Reviewer #2: Yes

3. Has the statistical analysis been performed
appropriately and rigorously? 

Reviewer #1: Yes

Reviewer #2: Yes

4. Have the authors made all data underlying the
findings in their manuscript fully available?

Reviewer #1: Yes

Reviewer #2: Yes

5. Is the manuscript presented in an intelligible
fashion and written in standard English?

Reviewer #1: Yes

Reviewer #2: Yes

6. Review Comments to the Author

Reviewer #1: In page 10, in the line "A multilevel model was then set up to assess
the relation-

ship between caste affiliation school performance for all six schools combined,

once again controlling for the above mentioned five covariates.", there is a
small

error. I believe the correct way to write this sentence will be:

A multilevel model was then set up to assess the relationship between caste

affiliation `AND' school performance for all six schools combined, once again

controlling for the above mentioned five covariates.

Reviewer #2: A couple of points might be considered:

1. While there is a detailed discussion of results on academic performance and its
relation to socio-economic status, there is no discussion on the two other measures,
self-esteem (SES) and satisfaction with life (SWLS). How were these related to
academic performance? Also, what were the gender differences in performance with
respect to their caste?

2. The Discussion needs to engage with Bourdieu, that is, how far it affirms the
theory of cultural/social capital.

Some sentences need polishing to improve the readability of the paper. For
instance:

Line in Introduction: 'Based on the yearly report of Ministry of Human Resource
Development (ESAG 2018), at Senior Secondary school, out of the 24.7 students
who

got enrolled, 5.9 were from the SC-ST category..." What are 24.7 and 5.9?
percentages?

The line in the Discussion 'Thus, the influence of social backwardness on
education

may be alleviated by better economic condition and worsened by economic
backwardness.' is confusing. It might either be modified or deleted.

The last line in the Discussion section: 'then it would imply that we may find a

greater gap in academic performance if we include students from rural schools who are
mostly from the

lower classes and who are less likely to study with students of higher class'. This
line is also not clear. "include" where?

7. PLOS authors have the option to publish the peer
review history of their article (what does this mean?). If published, this will
include your full peer review and any attached files.

If you choose “no”, your identity will remain anonymous but your review may still be
made public.

**Do you want your identity to be public for this peer review?** For
information about this choice, including consent withdrawal, please see our
Privacy Policy.

Reviewer #1: **Yes: **Anwesha Bandyopadhyay

Reviewer #2: **Yes: **Namrata Gupta

---

## [Author Response · Author response to Decision Letter 1]

14 Oct 2020

Dear Editors

On behalf of my fellow authors I herewith resubmit the manuscript “The Governmental
Ranking of Class and Academic Performance of Indian Adolescents” PONE-D-19-31769R1
for consideration in PLOS ONE. We have revised our manuscript substantially and
according to the reviewer’s sugges-tions. All authors listed in the title page have
read the manuscript, attest to the validity and legitimacy, and agree to its
submission to PLOS ONE. The authors report no financial interest or potential
conflict of interest.

We believe that our manuscript would perfectly fit in your prestigious journal and
thus we would be delighted to undergo your review process and are looking forward to
your valuable response.

On behalf of the authors, yours sincerely

Roshin Kunnel John and Jens Gaab

Reviewer #1:

In page 10, in the line "A multilevel model was then set up to assess the
relation-ship between caste affiliation school performance for all six schools
combined, once again controlling for the above mentioned five covariates.", there is
a small error. I believe the correct way to write this sentence will be: A
multilevel model was then set up to assess the relationship between caste
affiliation `AND' school performance for all six schools combined, once again
controlling for the above mentioned five covariates. 

Response: We have corrected the sentence accordingly. 

Reviewer #2: 

While there is a detailed discussion of results on academic performance and its
relation to socio-economic status, there is no discussion on the two other measures,
self-esteem (SES) and satisfaction with life (SWLS). How were these related to
academic performance? Also, what were the gender differences in performance with
respect to their caste?

Response: We have included the calculations and results for SES, SWLS as well as
gender, family income and age in the results section and also addressed these
findings in the discussion section with two added sentences. 

The Discussion needs to engage with Bourdieu, that is, how far it affirms the theory
of cultural/social capital.

Response: While we fully agree that our manuscript is possibly relevant to the area
of sociology, we must admit that none of the authors has any backgrpund or
experience in this discipline. Thus, any informed discussion of Bourdieu would
appear assumptive from our side. We hope that this is not a major problem, but of
course we hope that our paper and the results will be picked up by relevant
disciplines, such as sociology, educational sciences and policy studies. 

Line in Introduction: 'Based on the yearly report of Ministry of Human Resource
Development (ESAG 2018), at Senior Secondary school, out of the 24.7 students who
got enrolled, 5.9 were from the SC-ST category..." What are 24.7 and 5.9?
percentages?

Response: Yes, the % was missing and we have corrected the sentence accordingly. 

The line in the Discussion 'Thus, the influence of social backwardness on education
may be alleviated by better economic condition and worsened by economic
backwardness.' is confusing. It might either be modified or deleted.

Response: We deleted the sentence.

The last line in the Discussion section: 'then it would imply that we may find a
greater gap in academic performance if we include students from rural schools who
are mostly from the lower classes and who are less likely to study with students of
higher class'. This line is also not clear. "include" where?

Response: We deleted the sentence.

to reviewers.docx
---

## [Editor Report · Decision Letter 2]

16 Oct 2020

Governmental Ranking of Class and Academic Performance of Indian Adolescents

PONE-D-19-31769R2

Dear Dr. Gaab,

We’re pleased to inform you that your manuscript has been judged scientifically
suitable for publication and will be formally accepted for publication once it meets
all outstanding technical requirements.

Kind regards,

Tomáš Želinský, Ph.D.

Academic Editor

PLOS ONE

Additional Editor Comments (optional):

Once again I would like to thank you for your patience and addressing reviewers'
comments throughout the review process of your paper.
---

## [Editor Report · Acceptance letter]

21 Oct 2020

PONE-D-19-31769R2 

The governmental ranking of class and the academic performance of Indian adolescents 

Dear Dr. Gaab:

I'm pleased to inform you that your manuscript has been deemed suitable for
publication in PLOS ONE. Congratulations! Your manuscript is now with our production
department. 

Kind regards, 

on behalf of

Dr. Tomáš Želinský 

Academic Editor

PLOS ONE